# Impact of Optimizing Synthetic Image Similarity on Downstream Task Augmentation

Anonymous Full Paper
Submission 42

## Abstract

Using generative machine learning to generate synthetic medical data is an increasingly common method of augmenting limited datasets in segmentation and classification tasks. Typically, the quality of the data is measured by its similarity to the training data, as measured by the Fréchet Inception Distance (FID). In this paper we present three synthetic image selection algorithms that can be applied to GAN and Diffusion models after training with the aim of improving the quality of synthetic dataset and the downstream augmentation effectiveness. Our study shows that while the algorithms can consistently improve the FID significantly (up to a 27.38% reduction) in GAN generations, the results are mixed for diffusion models. Additionally, this improvement in FID has no significant impact on the downstream augmentation effectiveness of either model. This suggests that optimising the FID is not a good method for improving the augmentation efficacy of synthetic data.

## 1 Introduction

For effective machine learning we require large amounts of high quality data. In fields such as medical imaging, privacy concerns, costs, and pathology rarity can limit the amount of training data we have available [1]. To alleviate this challenge we can use generative machine learning models, such as Progressively Growing Generative Adversarial Networks [2] (PGGANs) or diffusion models[3, 4], to create synthetic data which can then be used to augment training tasks to improve performance [5–7], reduce bias [8, 9], and even address privacy concerns [10, 11]. Evidence suggests that the quality of this synthetic data, as measured by the Fréchet Inception Distance (FID), is predictive of how effectively it will augment medical imaging tasks [12]. In this paper we will present three novel synthetic image selection algorithms, which can be applied after training generative models, with the aim of improving synthetic image quality. We will first introduce each selection algorithm before evaluating their impact on image quality (section 4.1) and then on downstream machine learning tasks (section 4.2).

### 1.1 Machine Learning

Traditional machine learning can be summarised as a method by which we approximate a mapping function between two dataspaces:

$$f : \mathcal{X} \to \mathcal{Y}, \tag{1}$$

where $\mathcal{X}$ is the input—space, and $\mathcal{Y}$ is the output-space. Similarly to this, we can define a generative model as mapping between a latent—space $\mathcal{Z}$ and our typical input—space:

$$g : \mathcal{Z} \to \mathcal{X}, \tag{2}$$

or with conditional generation:

$$g : \mathcal{Z}_C \to \mathcal{X}, \tag{3}$$

where $\mathcal{Z}_C$ is the space of noise vectors appended with class labels (e.g. as one—hot encoded vectors). The main challenge in many machine learning tasks is the limited amount of training data $X_T \in \mathcal{X}$, $Y_T \in \mathcal{Y}$ which we rely on to approximate the spaces $\mathcal{X}$ and $\mathcal{Y}$. The more training data we have from a wide variety of sources, the better $X_T$ approximates the complete $\mathcal{X}$, and therefore the better our model will learn the mapping $f$. The goal of synthetic augmentations is to fill in some of the gaps in $X_T$ and improve our approximation of $\mathcal{X}$. Regions of $\mathcal{X}$ that are very sparsely represented in $X_T$, or simply not present, would be the most effective place to add new data. However, generative models are confined to the same regions of $\mathcal{X}$ as $X_T$, and so, in order to measure the synthetic data's quality we often use the Fréchet Inception Distance (FID) to measure synthetic image similarity, with lower FID representing more similar, and therefore better, synthetic images as they are more likely to inhabit regions of $\mathcal{X}$.

### 1.2 Fréchet Inception Distance

The Fréchet Inception Distance (FID) is defined as the Wasserstein-2 distance between the distribution of real ($\mathbb{P}_r$) and synthetic ($\mathbb{P}_s$) inception vectors, where each individual inception vector is the output of a pretrained inception network [13]:

$$FID(\mathbb{P}_r, \mathbb{P}_s) = |\mu_r - \mu_s|^2 + Tr(\Sigma_r + \Sigma_s - 2(\Sigma_r \Sigma_s)^{\frac{1}{2}}), \tag{4}$$

where $\mu_{r,s}$ are the respective means of the distributions and $\Sigma_{r,s}$ the covariances. Data which is very dissimilar will have a large FID, similar data a much lower FID with identical data returning an FID of zero. When evaluating conditional data it can often be informative to consider the FID of each class individually as well as their overall similarity.

## 2  Method

Using the Kermany dataset [14] we aim to augment a U-Net segmentation model for segmenting intraretinal fluid related to Diabetic Macular Edemas (DME). The U-Net will be evaluated using the Dice score[15]. The segmentation dataset had 750 images from 521 patients.

First, we trained a single PGGAN model for each task, then apply the various synthetic image selection algorithms to the same PGGAN model. We will then evaluate the improvements to image similarity as measured by the FID, then we will measure the model effectiveness on downstream segmentation tasks. In order to augment segmentation tasks we generate images with the grey-scale OCT scan in the red channel and the mask in the green channel. We can calculate the FID for both the full RGB image, as well as the OCT scan on its own.

With selected datasets we will then train our segmentation model repeatedly. Each model will be randomly initialised with the same hyper-parameters, trained, and tested at least thirty times for a given number of synthetic augmentation images, which we will increase gradually to build a curve relating the number of synthetic images to the model performance. U-Nets were trained using the Adam optimizer[16], a learning rate of $1 \times 10^{-3}$, a batch size of 5, and up to 100 epochs with early stopping, with binary cross entropy loss for the predicted masks.

To test the general applicability of these algorithms we will then perform a similar test with MNIST classification, where images are generated by a diffusion model. The classifier was a simple neural network with two hidden layers of size 50 and 25. Using an Adam optimizer with a learning rate of $5 \times 10^{-3}$, a batch size of 1000, and a mean squared error loss, with a maximum of 50 training epochs, typically finishing earlier using early stopping.

## 3  Synthetic Image Selection and Refinement Algorithms

Here we propose and test three algorithms for refining the quality of synthetic image datasets. The goal of these algorithms is to run efficiently, quickly, and reliably alongside a generator model which will generate a batch of synthetic images from which a subset will be selected. The process then repeats until we have the desired number of synthetic images. These individual selection algorithms can then be applied modularly, selecting from the outputs of another algorithm, allowing for their easy combination.

The first and third algorithms (Section 3.1 and Section 3.3 respectively) use the FID (or its inception model) to directly improve the FID of the selection, while the second (Section 3.2) focuses on features which are more specific to our observed poor generations and retinal OCT scans and is thus less generalisable.

### 3.1  FID Swapping Algorithm ($A_1$)

The FID Swapping Algorithm works by iteratively swapping images in and out of an 'in' group which will eventually be our selected subset. This process is shown in fig. 1. First, we randomly split the synthetic data into two lists, the 'in group' and 'out group', where every image is assigned a random weight. For images in the 'in group' this weight will represent the probability that it will be swapped with an image in the 'out group', for images in the 'out group' it will represent the probability that they will be randomly selected to swap with an image in the 'in group'. We calculate the FID before and after this swapping, keeping the proposed swap if the FID of the 'in group' after the swap is lower than the previour in'group', and reverting the swap if the FID is the same or higher. To help our model converge to the optimal 'in group' we will then update the weights of images such that good images should have a lower probability of being swapped out, and bad images a lower probability of being swapped in.

First, at step $n$, we will define a weighting factor $k$ which determines the strength of the update:

$$k = \frac{\|FID_n - FID_{n-1}\|^\alpha}{FID_0} \quad (5)$$

Where $FID_n$ is the FID after the proposed swap, $FID_{n-1}$ is the FID of the previous 'in group', $FID_0$ is the initial FID score and $\alpha$ is a hyper-parameter we typically set to 1.

After the swap has been accepted or rejected, images in the 'in group' that were part of the proposed swapping (and have now either just been moved in or have stayed) have their probability of being swapped updated according to:

$$P_i = (1 - k)P_i \quad (6)$$

Where subscript $i$ simply denotes each individual image which was swapped. Similarly we update the selection weights of the tested images in the 'out group' by:

$$W_i = (1 - k)W_i \quad (7)$$

After a swap the probability of selecting each of the images becomes their probability of being chosen

again to be swapped in. The effect of $k$ is that as the difference becomes larger the probabilities are multiplied by a smaller factor and thus are less likely to be chosen/swapped in the future.

In order to prevent stagnation as $P_i \to 0$, $\forall i$ we increase the probability and weight of those images which were not chosen in iteration $n$:

$$P_j = (1 - \frac{k}{2})P_j + \frac{k}{2} \qquad (8)$$

$$W_j = (1 - \frac{k}{2})W_j + \frac{k}{2} \qquad (9)$$

This eventually reaches an equilibrium in the number of images being swapped at each iteration with a gradual decrease in the FID of the in-group.

We then return this subgroup and repeat the algorithm until we have as many output images as we require.

## 3.2 Contour-Based Selection Algorithm ($A_2$)

Unlike the other algorithms presented in this paper the Contour-Based Selection Algorithm is designed specifically for generated retinal OCT images. While this limits the broad application it may provide useful insight into whether improving the FID is a worthwhile goal and whether we can rely on the tools of the FID in order to optimise our image selection.

One common issue in our synthetic retinal scans was the presence of large, blurry, artifacts which are not realistic and likely confuse classification and segmentation training. We detect images with these artifacts by measuring the length of algorithmically determined contours. These contours are found using the scikit-image package's find_contours function [17]. Once these are calculated, we reject images with above average mean contour length and accept those with below mean length. This process is repeated until we have enough outputs.

This process is shown in fig. 2. Examples of above and below average contour length data can be seen in fig. 3.

One significant flaw with this algorithm is that the images themselves may be of high quality but due to a different class of image have a larger contouring. For example in MRI brain scans different layers of the scan have different diameters and thus longer or shorter contours than the average. We will discuss this further in section 4.

## 3.3 GMM-Based Selection Algorithm ($A_3$)

The final of the three refinement algorithms presented here relies on the Inception network which is used within the calculation of the FID.

When calculating the FID, images are first passed as input to a pretrained Inception model and converted into 2,048 dimensional feature vectors. The FID is then calculated as the Wasserstein-2 distance between these clusters (See Section 1.2). To select images which are more similar to the training data we fit a Gaussian Mixture Model (GMM) [18] to our data and select images using the log-likelihood that they are from the original distribution. GMMs are not effective in high dimensions so first we embed both the synthetic and generated data using t-SNE dimension reduction before fitting a GMM to the data and scoring. The log-likelihood threshold varies depending on the overall quality of the dataset as worse datasets with higher FID will necessarily be further from the GMM fitted means of the training data. In general, we found that the most effective solution was to select images which have a log-likelihood score above the mean score of images with an above average score.

If the threshold is too high we will select a very small subset of data most likely in overrepresented classes which risks reinforcement of bias within the training data. Similarly, if the threshold is too low we risk including too many low-quality images into our training data. The overall process is shown in fig. 4.

# 4 Results

## 4.1 Improvements in Similarity

In table 1 we can see that the three algorithms and their combinations provide significant improvements to the dataset's FID. In general, using more algorithms provides a greater drop in FID with some variation depending on the order. In appendix B we can see the impact of these selection algorithms on multiple PGGANs of varying initial quality, showing that $A_1$ and $A_3$ provide robust improvements across a variety of PGGAN models while $A_2$ appears to be dependent on the initial model quality.

In table 2 we can see the results of applying these algorithms to a diffusion model which is generating MNIST digits. Like previous experiments, $A_3$ provides consistent improvements across all classes, though not as dramatically as before. $A_1$ and $A_2$ however are consistently making no difference, or making the model marginally worse. For $A_2$ this is likely due to the low resolution of MNIST, making it unsuitable. For $A_1$ however, the results are surprising given the algorithm is designed to always return a subgroup with lower FID than the original. The failure here might suggest that the algorithm was converging too quickly on a subgroup and that the hyperparameters require further tuning before being appropriate for use here. Similar to $A_2$ this may be a result of the low resolution altering the behaviour

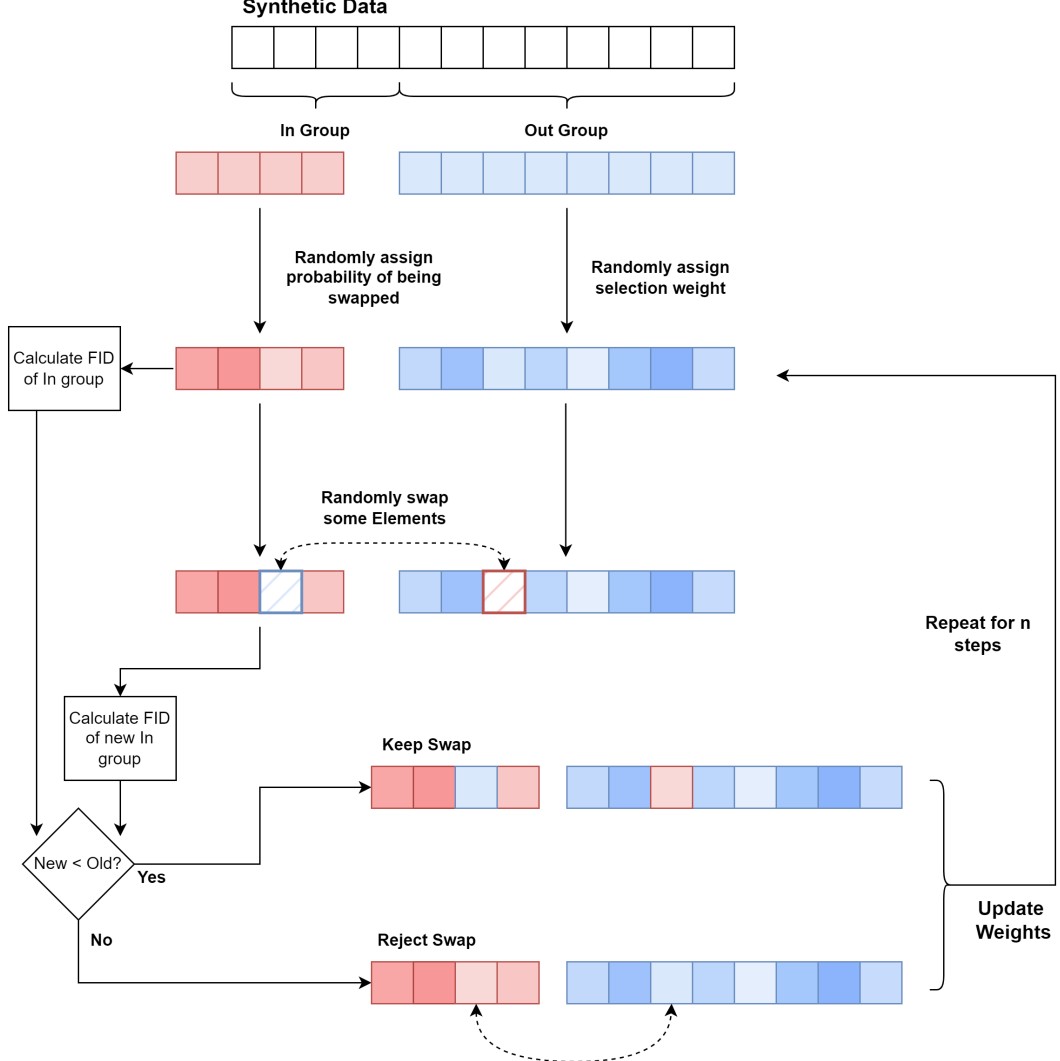

**Figure 1.** Flow chart showing the steps involved in the random image-swapping algorithm designed to gradually find the lowest FID sub-list of predetermined size from any presented list of synthetic images when compared to the training data. First images are randomly split into "in" and "out" lists where "in" refers to the list we are interested in optimising. According to these weights the images are swapped and the FID is tested. If the FID is lower the swap is kept and the weights are updated.

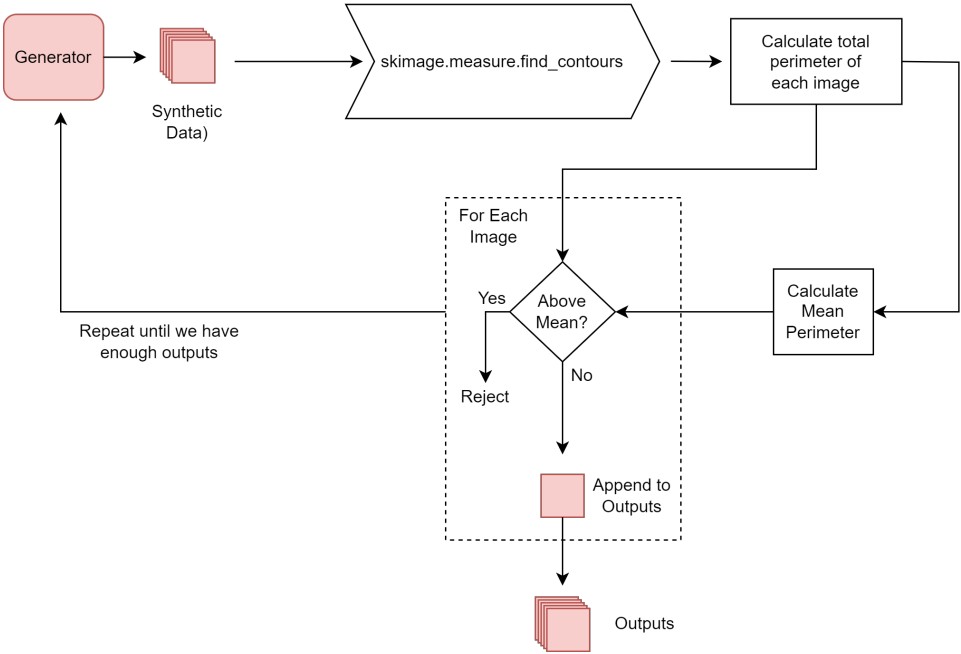

**Figure 2.** Flow chart showing the process of how synthetic retinal images are selected by first finding the contours of the scan then rejecting images that have an above average contour length. The process repeats until enough images have passed the threshold. This method is designed to ideally capture the most egregious of poor quality generations as opposed to directly improving the image similarity.

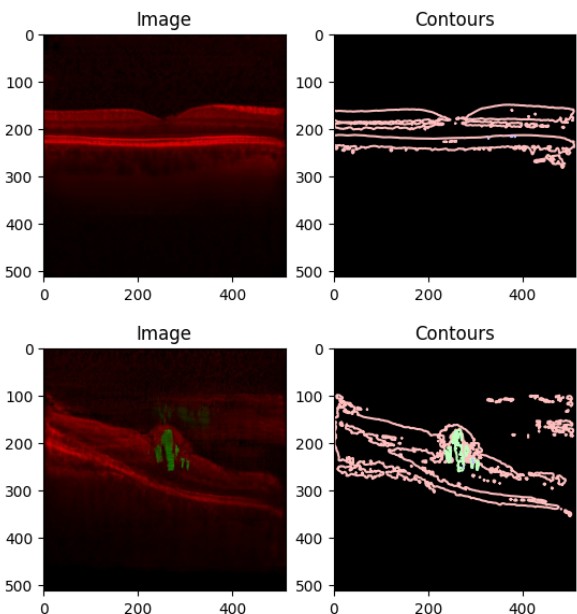

**Figure 3.** An example of a good quality image and it's contours (top row) and a similar example with poor quality image with significant artifacts (bottom row)

| Algorithm | FID ↓ | ΔFID | %ΔFID |
|-----------|-------|------|-------|
| None | 95.42 | - | - |
| $A_1$ | 86.02 | -9.402 | -9.854% |
| $A_2$ | 81.4 | -14.02 | -14.69% |
| $A_3$ | 76.51 | -18.91 | -19.82% |
| $A_5$ | 72.82 | -22.6 | -23.68% |
| $A_{12}$ | 76.08 | -19.34 | -20.27% |
| $A_{21}$ | 74.87 | -20.55 | -21.53% |
| $A_{13}$ | 71.89 | -23.53 | -24.66% |
| $A_{23}$ | 70.65 | -24.77 | -25.96% |
| $A_{31}$ | 68.15 | -27.27 | -28.58% |
| $A_{32}$ | 72.04 | -23.38 | -24.51% |
| $A_{123}$ | 69.04 | -26.38 | -27.64% |
| $A_{213}$ | **68.04** | **-27.38** | **-28.70**% |
| $A_{231}$ | 68.12 | -27.3 | -28.61% |
| $A_{132}$ | 70.23 | -25.19 | -26.39% |

**Table 1.** Table of FID improvements for $A_1$, $A_2$, $A_3$, and their combination. The combination notation is read as the leftmost algorithm sampling from the right i.e. $A_{mn}(G(z)) = A_m(A_n(G(z)))$. The change in FID and percentage change in FID are in reference to the unselected values presented as "None"

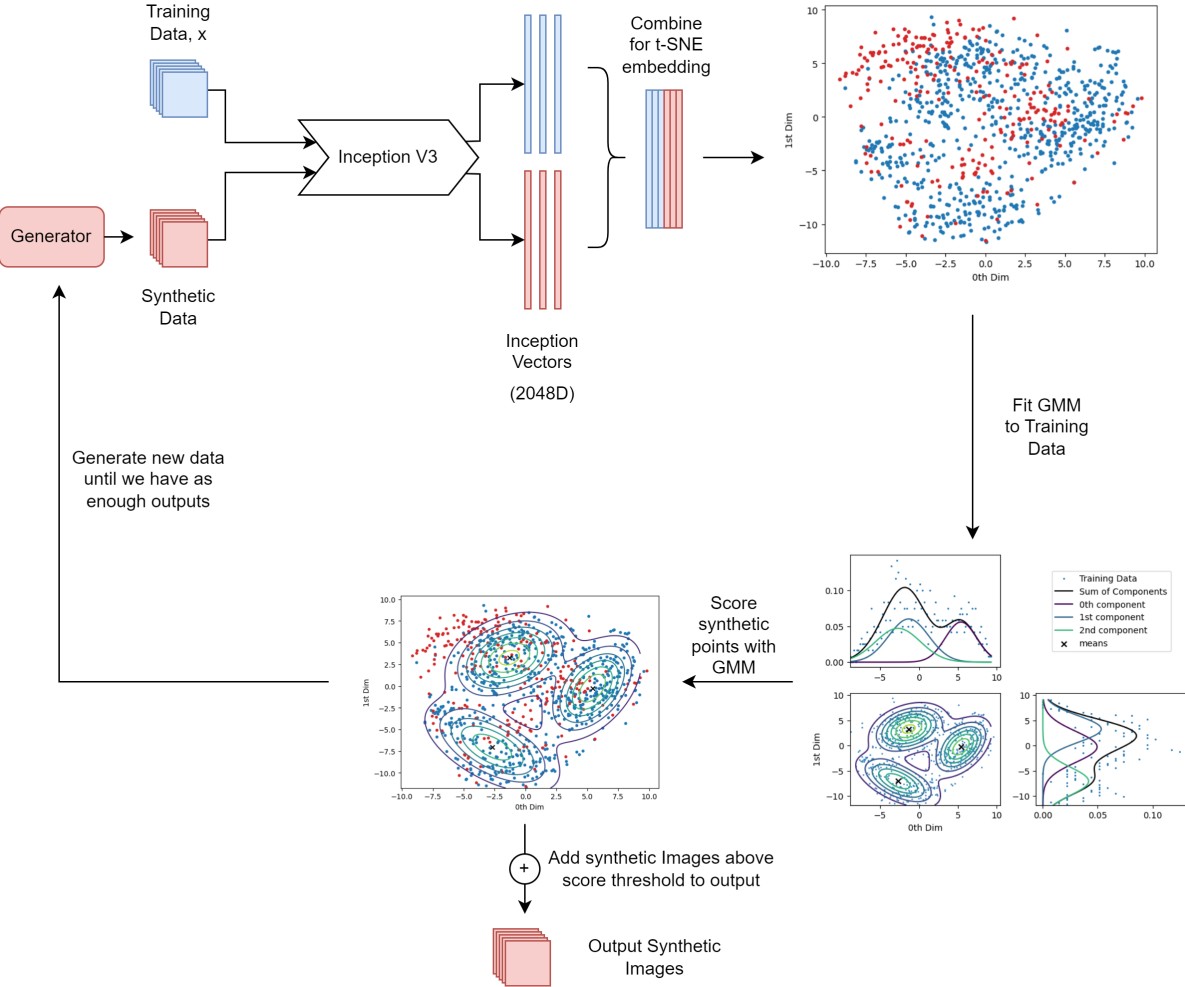

**Figure 4.** Diagram showing how synthetic images are scored and selected by fitting a Gaussian Mixture Model (GMM) to the t-SNE reduced inception vectors of the training data.

| Class | Unselected | $A_1$ | $A_2$ | $A_3$ |
|---|---|---|---|---|
| 0 | 127.12 | 127.91 | 127.09 | 122.69 |
| 1 | 129.43 | 133.04 | 145.49 | 125.71 |
| 2 | 145.66 | 150.36 | 147.61 | 139.69 |
| 3 | 133.38 | 133.41 | 130.78 | 124.26 |
| 4 | 137.20 | 136.05 | 139.06 | 129.30 |
| 5 | 128.29 | 140.77 | 129.85 | 124.80 |
| 6 | 129.67 | 135.96 | 133.97 | 123.01 |
| 7 | 139.88 | 142.25 | 148.45 | 132.41 |
| 8 | 118.70 | 134.04 | 115.85 | 111.86 |
| 9 | 123.27 | 124.17 | 125.59 | 118.05 |
| Mean | 131.26 | 135.80 | 134.38 | 125.18 |

**Table 2.** FID Improvements of selection algorithms $A_{1\rightarrow3}$ on the conditional outputs of an MNIST diffusion model.

of the inception model, but given the reasonably good performance of $A_3$ this is not considered likely.

## 4.2 Improvements to Downstream Tasks

### 4.2.1 Segmentation

In fig. 5 we can see the difference, or lack thereof, between the PGGAN's unselected outputs against those that have been selected by $A_{1\rightarrow3}$. We can clearly see that there is no significant difference between the selected, and unselected datasets. In fig. 6 we can see that this holds true for the best performing algorithm combinations. This suggests that the improvements to FID are not indicative of improvements to synthetic data's augmentation efficacy.

### 4.2.2 MNIST Classification

Similar to section 4.2.1 in fig. 7 we can see that there is no significant difference between unselected and selected datasets.

## 5 Conclusion

In this paper we have presented three seperate algorithms for selecting machine learning generated images to create a higher-quality dataset for augmentation. In section 4.1 we can see that the algorithms improve the FID by up to 27.38%. The GMM-based algorithm provides the most robust improvements across a variety of generative models while the algorithms $A_1$ and $A_2$ are less effective with an MNIST generating diffusion model. Results suggest that $A_2$ is limited by the resolution of the generations and the initial quality of the generative model, while $A_1$ may require further tuning to be effective with diffusion models, or simply might not be effective with this generative technique.

When applying these improved FID datasets to downstream segmentation and classification tasks, we saw no significant improvement in performance for these tasks regardless of the FID. One potential reason for this is data memorization. In appendix C we can see that the $\alpha$-precision changes as we apply selection algorithms, while $\beta$-recall stays incredibly low, suggesting memorization[19]. This suggests that while we might be improving the FID, the amount of novel information that is being used in augmentation is unaffected and is relatively low. Another potential consideration is that with the FID we only measure the similarity of the produced synthetic data, and not the assigned labels with which we train or how well the model is learning to generate data from or with labels. So, future selection algorithms might focus instead on measuring the effectiveness of the entire generative system, rather than just the inputs.

The results presented here suggest that the FID alone is not a suitable metric with which to optimise synthetic image datasets for augmentation, and a more comprehensive characterisation of the augmentation dataset should be considered.

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

## A Example Generations

$A_2$

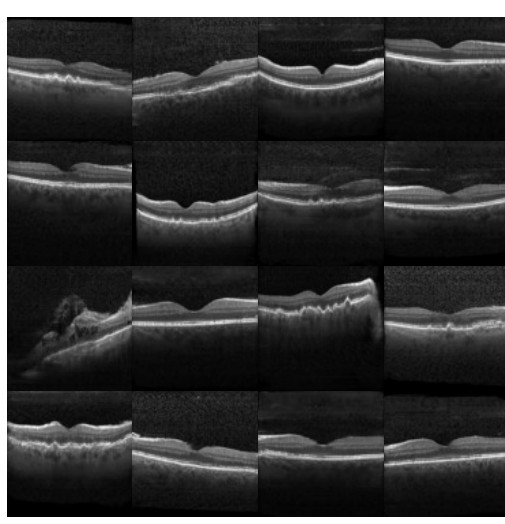

**Figure A.3.** Uncurated examples of Synthetic data refined using the contour-based selection algorithm presented in section 3.2.

Raw PGGAN Output

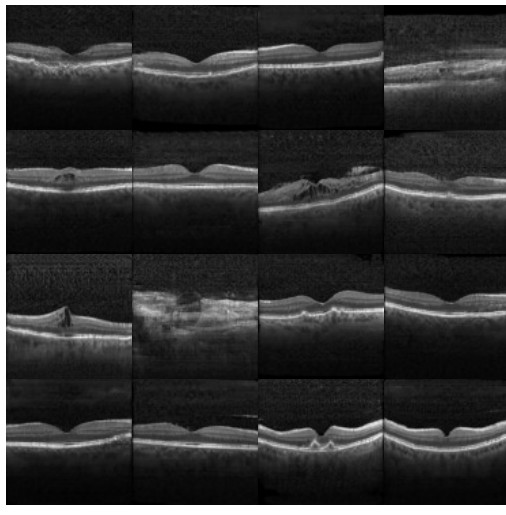

**Figure A.1.** Uncurated and unrefined PGGAN Generations of OCT retinal scans.

$A_3$

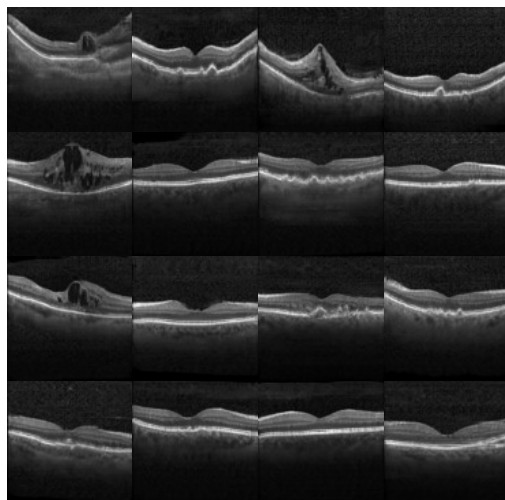

**Figure A.4.** Uncurated examples of synthetic data refined using the GMM-based selection algorithm presented in section 3.3.

## B Improvements to FID across Various PGGAN Models

In table B.1 we can see that these results are relatively consistent for a variety of PGGANs with different starting FIDs. The results in table B.1 are calculated using the entire RGB image while table 1 use only the greyscale image, hence giving the values in table B.1 a lower absolute value. One difference is in the $A_2$ results, which are not as consistent as

$A_1$

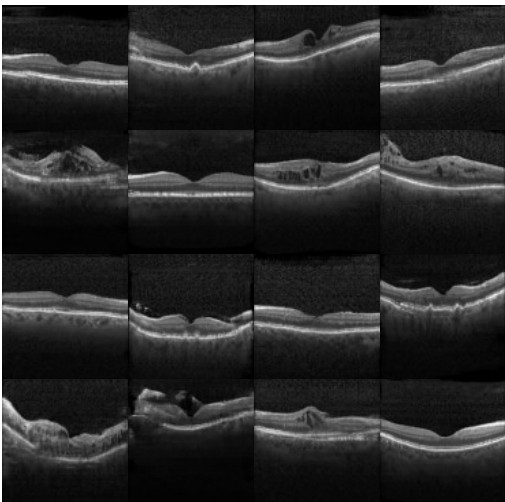

**Figure A.2.** Uncurated examples of synthetic data refined using the FID Swapping Algorithm presented in section 3.1

| Data Type | None | $A_1$ | $A_2$ | $A_3$ |
|-----------|------|-------|-------|-------|
| Retinal OCT | 70.15 | 57.99 | 64.25 | 55.89 |
| Retinal OCT | 205.69 | 185.99 | 212.44 | 197.19 |
| Retinal OCT | 72.66 | 64.49 | 76.40 | 68.97 |
| Retinal OCT | 34.12 | 28.61 | 33.21 | 26.96 |
| Retinal OCT | 105.32 | 91.48 | 98.06 | 96.32 |
| Retinal OCT | 33.04 | 28.82 | 33.93 | 29.91 |
| Retinal OCT | 35.48 | 27.76 | 32.31 | 26.89 |

**Table B.1.** Results of Selection Algorithms on a variety of Retinal OCT PGGANs.

$A_1$ and $A_3$, likely due to $A_1$ and $A_3$'s primary focus being the FID while $A_2$ focuses primarily on detecting large artifacts. If the PGGAN has a high initial FID these artifacts may be prevalent enough that detecting and removing the largest contours does not improve FID, but instead increases the imbalance of data, resulting in an increase in FID.

# C   Improvements   to   $\alpha-$ Precision and $\beta-$Recall

An alternative measure for measuring synthetic image quality is the $\alpha$—Precision and the $\beta$—Recall as proposed in [19]. $\alpha$—Precision can be defined as "'the fraction of synthetic samples that resemble the "most typical" fraction $\alpha$" of the dataset while $\beta$— Recall is "the fraction of real samples covered by the most typical fraction $\beta$ of synthetic samples". The optimal value is a straight line with gradient one. In fig. C.1 and fig. C.2 we can see the changes to these values after using our selection algorithms to our retinal and MNIST generations respectively. The results in fig. C.1 suggest that while our synthetic data is made to look more "realistic" (improved $\alpha$—Precision) after selection, there is a fundamental issue with memorization which leaves the $\beta$—Recall unaffected as this memorization in the generative model cannot be undone via selection. In fig. C.2 we see similar results, but with a much worse initial value.

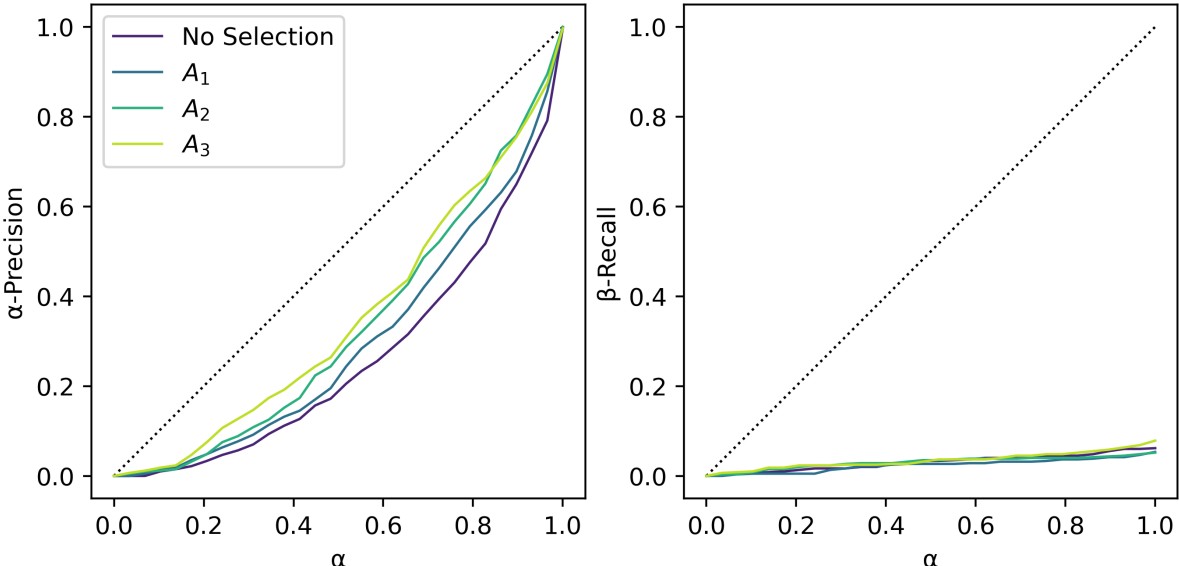

**Figure C.1.** Alpha Precision and Beta Recall for selected and unselected retinal image datasets.

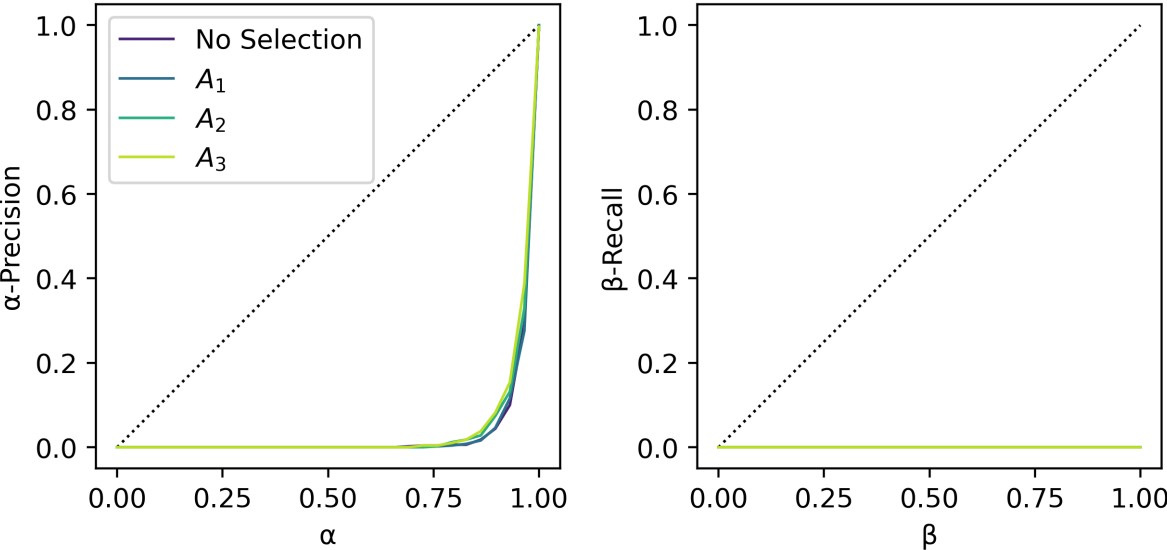

**Figure C.2.** Alpha Precision and Beta Recall for selected and unselected MNIST image datasets.

