# OpenReview forum: "Impact of Optimizing Synthetic Image Similarity on  Downstream Task Augmentation"
_NLDL.org/2026/Conference — Submitted to NLDL 2026_

### Official Review · Reviewer_o9rv · 2025-09-22
**Interesting concept, but limited evidence and unclear benefit from (FID-based) OCT augmentation**

**Rating:** 2
**Confidence:** 4
**Final Rating:** 2
**Final Confidence:** 4

**Summary:**

The paper examines data augmentation for optical coherence tomography (OCT) using generative image models. Some results in the literature indicate the quality as measured by FID can translate to improved performance. This paper shows that FID can be improved by using selection algorithms on top of a genrative model, but that it does not translate to better downstream performance for a segmentation task. Specifically, the paper presents three algorithms for image selection, one of which is specially designed to be relevant for OCT scan images.

**Strengths:**

The use of data augmentation in this type of task is of practical importance, and finding better ways to do this using generative models is very interesting.

Although results are negative for the particular selection algorithms examined, it could be of interest to the community to better understand if and how FID or other measure correlate with downstream segmentation performance. It is not clear to me if FID is used as a criterion by practitioners in this field for designing augmentation schemes, but if that is the case, this paper provides some evidence that this is not always beneficial.

The paper is quite readable, apart from minor typographical issues, and it it easy to understand the general setting. Results are presented in a nice and fair manner, and good figures are included to more clearly communicate the algorithms.

**Weaknesses:**

Results are very limited, and includes only a demonstration on MNIST as well as one more realistic OCT task. The focus is on showing how selection of which images to include from a generative model for augmenting the data affects performance, but since neither FID nor the coutour based algorithm seem to lead to an effective criterion, it is not clear what exactly can be concluded from the experiments. It would have been much stronger if some criterion could be identified that lead to improved performance.

A key issue addressed in section 3 is a combinatorial optimization problem. There are several well known algorithms for this, such as genetic algorithms, simulated annealing, and continuous relaxations. It would be useful with a general discussion of this and perhaps comparison with some of these more standard algorithms.

Section 3.1: Algorithm 1 is presented without much motivation for the particular choices. Does this algorithm correspond to some well-known combinatorial optimization algorithm?

Section 3.2: This is a novel contributions, and could perhaps benefit from a more detailed discussion.

I did not find a direct comparison between computing the FID on OCT vs OCT+mask as discussied in line 108.

Overall, the performance on the downstream taks does not appear to be affected positively by the image selection algorithms, but there is a clear improvement from the data augmentation. It would be nice with a more detailed discussion or analysis that looked into why this works, and what aspects of the augmentations that are beneficial.

There are multiple minor issues that should be addressed if the paper is published, listed below.

Minor issues
- Several times em-dash is used where it should by a hyphen.
- Line 54+57+85: There should not be an indent (new paragraph) here.
- Line 71: Although generative methods are restricted by their traning set, they do generalize, so perhaps this statement is a bit too strong.
- Line 75: The argument that more similar images are better is not clear to me - don't we ideally want images that are dissimilar but "realistic" in some sense?
- Please include a short description of the Kermany dataset.
- Describe or provide a reference to PGGAN
- Please include a brief outline of the neural network architectures used - perhaps in the supplement.
- The abbreviation OCT is not described
- The abbreviation DSC is not described
- Line 118: Missing superscript
- In table 1 and 2, how many images are retained for each setting?

**Final Justification:**

Insufficient experimental validation

**Justification:**

My justifications for recommending rejection at this point are

- The experiments are very limited and with no benefit demonstrated. This makes it unclear what exactly can be concluded from the study.
- The methods that are developed should be more clearly related to or contrasted with existing methods in the literature, e.g. methods for combinatorial optimization.

---

> ### Author Rebuttal · Authors · 2025-10-21
>
> Firstly, we would like to thank the reviewer for their valuable comments and insights. We hope that here we can adequately address their concerns or incorporate some of their feedback here. In this rebuttal I will try to structure it similarly to the "Weaknesses" section, responding to each new paragraph in order, for ease of navigation.
>
> - While we agree that the first two algorithms provide no clear criterion, the conclusion presented is that the FID alone is not an informative enough metric for discussion whether one dataset is better than another in augmentation tasks. However, as you have said, the tests are primarily focussed on one OCT task and a brief experiment with MNIST, which limits our ability to claim that our findings are true for all models, tasks, and datasets. To make this clear, we propose to re-emphasise (and partially reframe in the title and introduction) that our tests are specific to a small dataset problem (OCT segmentation) and that while the conclusions might indicate a broader trend, they do not provide a general conclusion.
>
> - This is an interesting idea, and we agree that the inclusion of these more traditional algorithms, and that a comparison, may prove useful and informative. However, to do so is beyond the scope of this submitted paper, beyond a brief discussion that these may also derive useful selection algorithms.
>
> - The design choices for Algorithm 1 are primarily made through experimental results and certainly has room for refinement. The goal of this algorithm's design was to make a robust algorithm that would always return a better or equal FID than the original generative model, and this aim naturally led to the design of the algorithm, rather than being based on another combinatorial optimization algorithm. This could be included in Section 3.1 to make the choices and motivation clearer, as well as highlighting the potential for improvements and alternatives derived from traditional algorithms.
>
> - A more detailed discussion of A2 can be added
>
> - Apologies, the FID on OCT vs OCT+mask is unclear, and the OCT+mask is only used in the appendix, adding to the confusion. To address this we will simply change all FID values to be calculated on the OCT scan as this makes the models more directly comparable. This is mostly a small oversight.
>
> - Since submitting this paper we have run such analyses and found that the synthetic augmentations improve segmentation by effectively reducing the number of false positives and false negatives compared to the unaugmented data, suggesting that the utility in this dataset is in the generated images without masks and not generated images with masks. These results can be included in the paper to better explain why we see the results we see, and thus, how they might be applicable, or not, to wider issues.
>
> - Thank you for including these minor issues, these don't need any response as they simply need fixed. However, we would like to thank you for taking the time to point them out to us.

---

### Official Review · Reviewer_6TiN · 2025-10-01
**Review of Manuscript**

**Rating:** 2
**Confidence:** 4

**Summary:**

In classification tasks, the Fréchet Inception Distance (FID) is a common metric for evaluating the quality of synthetic images. The authors of this paper challenge the appropriateness of FID for that purpose. They present three algorithms that generate synthetic images with lower FID scores, yet the resulting augmentation does not lead to a noticeable improvement in downstream tasks.

**Strengths:**

* This paper asks a fundamental question in the synthetic image generation domain.
* The three proposed methods for improving FID are well documented and sound. Experimental results show that FSD can improve by as much as 27%
* The experimental results also verify the claim made by the author in most cases that downstream task performance has not improved

**Weaknesses:**

* There is room for improving the details provided in the experimental section More details about how the results are achieved would help with reproducibility as well.
* Further experiments with more datasets will also be useful to cement the claim made within this paper.
* Providing some alternative to FID for such tasks will also strengthen the value of this paper.

**Justification:**

I appreciate the authors asking a fundamental question in this paper. But I believe the experiments provided in this paper are not sufficient to justify the claim. More thorough and rigorous experiments, involving more datasets, classifier and downstream tasks, are required to back the claim made in this paper.

---

> ### Author Rebuttal · Authors · 2025-10-21
>
> Firstly, we would like to thank the reviewer for their valuable comments and insights. We hope that here we can adequately address their concerns or incorporate some of their feedback here. In this rebuttal I will try to structure it similarly to the bullet-points in the  "Weaknesses" section for ease of navigation.
>
> - To address this we can add more details on our model to Section 2, primarily on the U-Net's size.
> - The title, abstract, and conclusion of this paper do potentially overstate the claim made within the paper, as you have correctly pointed out, to make this clear we propose to alter these to reemphasise that these tests are for a limited dataset OCT problem which we are trying to improve, with a brief MNIST experiment as an initial test into the generalisability of our results. We want to emphasise that this is not sufficient to claim that our findings hold true for all generative models, tasks, and datasets.
> - In Appendix C we provide alpha-precision and beta-recall as alternatives to the FID to provide further context. However, given the common use of the FID in synthetic data, we thought it would be the most appropriate metric to show.

---

### Official Review · Reviewer_TkBz · 2025-10-07
**Broadline Accept**

**Rating:** 4
**Confidence:** 3
**Final Rating:** 4
**Final Confidence:** 3

**Summary:**

The paper investigates whether optimizing the similarity metric FID between synthetic and real images can improve the effectiveness of data augmentation in downstream tasks. The authors propose three post-generation image selection algorithms — FID Swapping, Contour-Based, and GMM-Based methods — and evaluate them on medical image segmentation and MNIST classification. The results show that while these methods can significantly reduce FID, they do not lead to better downstream performance, suggesting that optimizing FID does not effectively enhance augmentation utility.

**Strengths:**

- The paper investigates the relationship between the generative image similarity metric (FID) and downstream task performance, focusing on the practical utility of synthetic data — an important and underexplored topic in generative modeling research.
- The experiments cover two main-stream generative models (GAN and Diffusion) and task types (medical image segmentation and general image classification), providing a well-rounded and representative evaluation.
- The three image selection algorithms (FID Swapping, Contour-Based, and GMM-Based) are clearly motivated and thoroughly described. The detailed experimental setup supports reproducibility and fair comparison.
- The paper provides a reflective discussion on model memorization and the limitations of FID, introducing α-Precision and β-Recall as complementary measures. The analysis is insightful and deepens understanding of the observed results.
- The paper is well-structured and clearly written, with well-designed figures and tables that effectively illustrate the algorithms and experimental findings.

**Weaknesses:**

- The paper primarily employs relatively small generative models and does not include large-scale or state-of-the-art models such as Stable Diffusion, FLUX, or DiT, nor their fine-tuned variants. This omission limits the generality and practical relevance of the conclusions.
- The A2 rule-based approach is tightly coupled with specific tasks (e.g., medical image segmentation) and lacks portability to other domains or data types, which restricts its broader applicability.

**Final Justification:**

Thanks for the rebuttal. It clarified some of the concerns. I’ll keep my score unchanged.

**Justification:**

The paper provides a careful and transparent empirical investigation of how optimizing FID relates to downstream task performance. While the scope of models and datasets is somewhat limited and one method lacks generality, the study is methodologically sound, clearly written, and offers useful evidence that challenges common assumptions about synthetic data evaluation. Overall, it represents a modest but meaningful contribution that merits weak acceptance.

---

> ### Author Rebuttal · Authors · 2025-10-21
>
> Firstly, we would like to thank the reviewer for their valuable comments and insights. We hope that here we can adequately address their concerns or incorporate some of their feedback here. In this rebuttal I will try to structure it similarly to the bullet-points in the  "Weaknesses" section for ease of navigation.
>
> - This is a fair point, the lack of general applicability of our conclusions to other, untested, generative models will be further emphasised and the problem more carefully framed to avoid confusion.
>
> - We agree. While we have some subsequent results (which are not in this paper and therefore not entirely relevant, but still interesting) suggesting the A2 algorithm has some broader utility for non-OCT tasks, the algorithm is designed to be specific to the OCT dataset used, and so we will emphasise this further.

---

### Official Review · Reviewer_dKRN · 2025-10-09

**Rating:** 2
**Confidence:** 4
**Final Rating:** 2
**Final Confidence:** 4

**Summary:**

This paper investigates the impact of optimizing synthetic image quality on the performance of downstream machine learning tasks. The authors propose three post-hoc selection algorithms designed to refine datasets generated by GANs and diffusion models by improving their Fréchet Inception Distance (FID), a common metric for image similarity. The methods are tested on a retinal OCT segmentation task and an MNIST classification task. The study finds that while the proposed algorithms can significantly reduce the FID of the synthetic datasets, this improvement does not translate into any significant performance gain on the downstream tasks. The authors conclude that FID alone may not be a suitable metric for optimizing the augmentation efficacy of synthetic data.

**Strengths:**

1) Relevant Research Question: The paper addresses a critical and practical question in generative modeling: Does improving a standard similarity metric like FID necessarily lead to better utility in downstream applications? The negative result presented contributes to an important discussion on how to evaluate generative models for practical use.

2) Novelty in Proposed Methods: The paper introduces three distinct algorithms for refining synthetic datasets

3) Clear Conclusion: The authors are direct about their negative findings and appropriately question the underlying assumption that lower FID equals better augmentation data.

**Weaknesses:**

The paper suffers from several significant weaknesses:

*Extremely Limited Experimental Validation*:

- The conclusions are drawn from a very narrow set of experiments, using only one medical imaging dataset (retinal OCT) and one simple computer vision dataset (MNIST). This lack of diversity makes it difficult to generalize the findings to other domains or data modalities.

- The study is limited to one PGGAN and one diffusion model. The results cannot be assumed to hold for other, more advanced generative architectures.

- The experimental settings are questionable. For instance, using a batch size of 5 for the U-Net segmentation task is unusually small and may lead to unstable training, potentially masking any real performance improvements from the refined datasets.

*Significant Drawbacks in Proposed Methods*:

- The premise of using FID for selection is flawed when generated images contain visible artifacts or structural differences not present in the training data (of the Inception network), as is common. Optimizing a distributional similarity metric may not be meaningful in such cases.

- The Contour-Based selection method (A2) is explicitly tailored for the retinal OCT scans and is acknowledged by the authors to be "less generalisable," limiting its overall contribution.

- The paper fails to establish the quality of the base generative models. If the generators were already poorly performing (e.g., suffering from mode collapse or memorization, as suggested by the very low β-Recall scores ), then no post-hoc selection method could create a genuinely useful dataset. The algorithms might simply be selecting the least flawed images from a poor-quality set.


*Underwhelming and Poorly Contextualized Conclusion*:

- The main result is that the proposed methods fail to improve downstream performance. While negative results are welcome, the numerous experimental weaknesses prevent this from being a convincing or generalizable conclusion. It is unclear if the failure lies with the principle of FID optimization or with this specific, limited implementation.

- The paper's conclusion clashes with a vast body of machine learning literature that has successfully demonstrated performance improvements from synthetic data augmentation. The authors fail to adequately contextualize their findings within this established work.


*Conceptual Oversimplification* (writing style):

The introduction's characterization of machine learning as merely a function mapping between "dataspaces" is reductive and inaccurate, suggesting a superficial framing of the research problem.

**Final Justification:**

I thank the authors for their rebuttal. As my major concerns have not been resolved, I decided to keep my score unchanged.

**Justification:**

My decision to reject this paper is justified by the combination of significant, unaddressed weaknesses.

The paper makes a strong claim: that optimizing for FID is not a good method for improving the augmentation efficacy of synthetic data. However, this claim is built upon a foundation of insufficient and potentially flawed experimental evidence. The lack of diversity in datasets and models, coupled with questionable training parameters, severely limits the generalizability and credibility of the results.

Furthermore, the proposed methods have inherent limitations, with one being task-specific and the others relying on a metric (FID) that may be inappropriate for the very data they are trying to improve. The paper does not adequately verify the quality of the underlying generative models, making it impossible to discern whether the failure is due to the selection principle or the poor quality of the initial data.

Given that the paper's primary contribution is a negative result, the authors needed to demonstrate through rigorous and extensive experimentation that their findings are robust and widely applicable.

---

> ### Author Rebuttal · Authors · 2025-10-21
>
> Firstly, we would like to thank the reviewer for their valuable comments and insights. We hope that here we can adequately address their concerns or incorporate some of their feedback here. In this rebuttal I will try to structure it similarly to the bullet-points in the  "Weaknesses" section for ease of navigation.
>
> *Extremely Limited Experimental Validation*
> - The lack of generalisability is a valid concern, and we agree that the narrow scope is not reflected in the title or in parts of the introduction. We propose that we would change the title and abstract to reflect the specific application focus of our work.
> - We will add this to the conclusion/discussion as we agree this is a vital limitation of the experiment that should be more clearly communicated.
> - The small batch size is the best found for this particular dataset and model when training without synthetic data. Potentially, when adding more synthetic data the batch size could be increased, but for this work we wanted to ensure that the only change in the training setup was the inclusion of synthetic data (selected or unselected). This potential flaw will be added to a discussion section to highlight the concerns you have raised here.
>
> *Significant Drawbacks in Proposed Methods*
> - This is a fair concern. Given the common use of the FID as a metric for synthetic data, it felt appropriate to test whether its optimisation was a valid goal (on its own at least). We propose to include this within a discussion section that highlights the inherent limitations of the FID.
> - Agreed. We will further highlight this in the abstract and conclusion.
> - The quality of the base models can be seen from the uncurated samples in figure A1, the base/unselected FIDs in Tables 1 and 2, and the alpha-precision and beta-recall scores in Figures C1 and C2. We agree that the algorithms may simply be selected the best of a bad batch, however, in the case of a very limited dataset where generated images are almost certain to have some flaws, this could still be valuable. That said, we agree that this should be discussed and emphasised more strongly.
>
> *Underwhelming and Poorly Contextualized Conclusion*
> - We agree that this is not strong enough for a general conclusion and did not wish for it to be interpreted as such. To address this, we propose to more clearly frame this as an experiment focussing primarily on one limited dataset and application, with a brief experiment into MNIST to test how effective our proposed solutions are on a non-OCT dataset. As you have pointed out, this has not been made clear enough in the body of the text and so slight changes are required to make the scope of our experiments and findings as clear as possible.
> - The paper demonstrates improvements from synthetic data augmentation (see figures 5, 6, and 7), in agreement with the literature. What we did not find were any further improvements to model performance when augmenting with selected synthetic datasets compared to unselected synthetic datasets. This, as far as we know, does not contradict current findings.
>
> *Conceptual Oversimplification*
> - While we find this framing can be useful for the broad consideration of machine learning, it is a simple model , and can be removed from the introduction for a more specific introduction to our problem, which is in line with our above proposals to highlight the specific scope of this work.

---

### Meta-Review · Area_Chair_raWj · 2025-10-31

**Recommendation:** Reject
**Confidence:** 4

**Metareview:**

This paper examines whether optimizing synthetic image similarity leads to improved downstream performance. The topic is clearly of interest and within scope for the conference.

All reviewers appreciated the paper’s clarity and honesty in reporting negative results, as well as its practical focus on a key but often ignored question in generative AI for medical imaging. However, several concerns were raised with one being prominent across reviews: the experimental scope is very narrow (limited datasets, small models). The authors’ rebuttal effectively clarified scope limitations and proposed reframing the paper to emphasize its exploratory nature, but did not add new empirical evidence, which is very limited in its current form to extract meaningful conclusions from the work (which themselves need work). However, there is clear scope to expand and improve on this work in the future.

---

### Decision · Program_Chairs · 2025-11-05

**Decision:**

Reject

**Comment:**

Based on the reviewers and AC comments, the paper cannot be presented at the conference.